# Spectral-Spatial Joint Classification of Hyperspectral Image Based on Broad Learning System

**Guixin Zhao** [1,2], **Xuesong Wang** [1], **Yi Kong** [1] and **Yuhu Cheng** [1,*]

1 School of Information and Control Engineering, China University of Mining and Technology, Xuzhou 221116, China; zgx@qlu.edu.cn (G.Z.); wangxuesong@cumt.edu.cn (X.W.); kongyi@cumt.edu.cn (Y.K.)
2 School of Computer Science and Technology, Qilu University of Technology (Shandong Academy of Sciences), Jinan 250353, China
* Correspondence: yhch@cumt.edu.cn

**Abstract:** At present many researchers pay attention to a combination of spectral features and spatial features to enhance hyperspectral image (HSI) classification accuracy. However, the spatial features in some methods are utilized insufficiently. In order to further improve the performance of HSI classification, the spectral-spatial joint classification of HSI based on the broad learning system (BLS) (SSBLS) method was proposed in this paper; it consists of three parts. Firstly, the Gaussian filter is adopted to smooth each band of the original spectra based on the spatial information to remove the noise. Secondly, the test sample's labels can be obtained using the optimal BLS classification model trained with the spectral features smoothed by the Gaussian filter. At last, the guided filter is performed to correct the BLS classification results based on the spatial contextual information for improving the classification accuracy. Experiment results on the three real HSI datasets demonstrate that the mean overall accuracies (OAs) of ten experiments are 99.83% on the Indian Pines dataset, 99.96% on the Salinas dataset, and 99.49% on the Pavia University dataset. Compared with other methods, the proposed method in the paper has the best performance.

**Keywords:** hyperspectral image; classification; Gaussian filter; broad learning system; guided filter

## 1. Introduction

Hyperspectral images (HSI) are widely used in various fields [1–4] due to their many characteristics, such as spectral imaging with high resolution, unity of spectral image and spatial image, and rapid non-destructive testing. One of the important tasks of HSI applications is HSI classification. At first, researchers only utilized spectral features for classification because the spectral information is easily affected by some factors, for example, light, noise, and sensors. The phenomenon of "same matter with the different spectrum and the same spectrum with distinct matter" often appears. It increases the difficulty of object recognition and seriously reduces the accuracy of classification. Then researchers began to combine spectral characteristics and spatial features to improve the classification accuracy.

The spectral feature extraction of HSI can be realized by unsupervised [5,6], supervised [7,8], and semi-supervised methods [7,9,10]. Representative unsupervised methods include principal component analysis (PCA) [11], independent component analysis (ICA) [12], and locality preserving projections (LPP) [13]. Some well-known unsupervised feature extraction methods are based on PCA and ICA. The foundation of some supervised feature extraction techniques for HSIs [14,15] is the well-known linear discriminant analysis (LDA). Many semi-supervised methods of spectral feature extraction often combine supervised and unsupervised methods to classify HSIs using limited labeled samples and unlabeled samples. For example, Cai et al. [16] proposed the semi-supervised discriminant analysis (SDA), which adopts the graph Laplacian-based regularization constraint in LDA

to capture the local manifold features from unlabeled samples and avoid overfitting while the labeled samples are lacking. Sugiyama et al. [17] introduced the method of the semi-supervised local fisher discriminant analysis (SELF). It consists of a supervised method (local fisher discriminant analysis [8]) and PCA. These feature extraction methods (PCA, ICA, and LDA) cannot describe the complex spectral features structure of HSIs. LPP, which is essentially a linear model of Laplacian feature mapping, can describe the nonlinear manifold structure of data and is widely used in the spectral feature extraction of HSIs [18,19]. He et al. [20] applied multiscale super-pixel-wise LPP to HSI classification. Deng et al. [21] proposed the tensor locality preserving projection (TLPP) algorithm to reduce the dimensionality of HSI. However, in LPP, it is difficult to fix the value of the quantity of nearest neighbors used to construct the adjacency graph [22]. The above spectral feature extraction methods are all realized by the dimensionality reduction, which results in losing some spectral information. The Gaussian filter [23] can smooth the spectral information without reduction of bands in order to remove the noise from HSI data. Because of the advantage of removing the noise from data and liner calculation characteristic, the Gaussian filter is widely used in the classification of HSIs [24,25]. Tu et al. [26] used the Gaussian pyramid to capture the features of different scales by stepwise filtering and down-sampling. Shao et al. [27] utilized the Gaussian filter to fit the trigger and echo signal waveforms for coal/rock classification. The spectra of four-type coal/rock specimens are captured by the 91-channel hyperspectral light detection and ranging (LiDAR) (HSL).

In terms of spatial feature extraction, a Markov model was initially adopted to capture spatial features [28]. However, it has two disadvantages, which are intractable computational problems and no enough samples to describe the desired object. Then the morphological profile (MP) model [29] was put forward. Even if MP has a strong ability to extract spatial features, it cannot achieve the flexible structuring element (SE) shape, the ability to characterize the information about the region's grey-level features, and the less computational complexity [30]. Benediktsson et al. [31] proposed the extended morphological profile (EMP) to classify the HSI with high spatial resolution from urban areas. In order to solve the problems of MP, Mura et al. [32] proposed morphology attribute profile (MAP) as a promotion of MP. The extended morphological profiles with partial reconstruction (EMPP) [33] were introduced to achieve the classification of high-resolution HSIs in urban areas. Subsequently, the extended morphological attribute profiles (EMAP) [34] was adopted to cut down the redundancy of MAP. The framework of morphological attribute profiles with partial reconstruction [35] had gained better performance on the classification of high-resolution HSIs. Geiss et al. [36] proposed the method of object-based MPs to get a great improvement in terms of classification accuracy compared with standard MPs. Samat et al. [37] used the extra-trees and maximally stable extremal region-guided morphological profile (MSER_MP) to achieve the ideal classification effect. The broad learning system (BLS) classification architecture based on LPP and local binary pattern (LBP)(LPP_LBP_BLS) [19] was proposed to gain the high-precision classification. However, LBP only uses the local features of pixels and needs to use an adjacency matrix, which requires a lot of calculation. In recent years, the guided filter [38–42] has attracted much interest from many researchers due to its low computational complexity and edge-preserving ability. The hierarchical guidance filtering-based ensemble classification for hyperspectral images (HiFi-We) [42] was proposed. The method obtains individual learners by spectral-spatial joint features generated from different scales. The ensemble model, that is, the hierarchical guidance filtering (HGF) and matrix of spectral angle distance (mSAD), can be achieved via a weighted ensemble strategy.

Researchers have paid a great deal of work to build various classifiers for improving the classification accuracy of HSIs [43], such as random forests [44], neural networks [45], support vector machines (SVM) [46,47], and deep learning [48], reinforcement learning [49], and broad learning systems [50]. Among these classifiers, the BLS classifier [51,52] has attached more and more research attention due to the advantage of its simple structure, few training parameters, and fast training process. Ye et al. [53] proposed a novel regularization

deep cascade broad learning system (DCBLS) method to apply to the large-scale data. The method is successful in image denoising. The discriminative locality preserving broad learning system (DPBLS) [54] was utilized to capture the manifold structure between neighbor pixels of hyperspectral images. Wang et al. [55] proposed the HSI classification method based on domain adaptation broad learning (DABL) to solve the limitation or absence of the available labeled samples. Kong et al. [56] proposed a semi-supervised BLS (SBLS). It first used the HGF to preprocess HSI data, then the class-probability structure (CP), and the BLS to classify. It achieved the semi-supervised classification of small samples.

In order to make full use of the spectral-spatial joint features for improving the HSI classification performance, we put forward the method of SSBLS. It incorporates three parts. First, the Gaussian filter is used to smooth spectral features on each band of the original HSI based on the spatial information for removing the noise. The inherent spectral characteristics of pixels are extracted. The first fusion of spectral information and spatial information is realized. Second, inputting the pixel vector of spectral-spatial joint features into the BLS, BLS extracts the sparse and compact features through a random weight matrix fine-turned by a sparse auto encoder for predicting the labels of test samples. The initial probability maps are constructed. In the last step, a guided filter corrects the initial probability maps under the guidance of a grey-scale image, which is obtained by reducing the spectral dimensionality of the original HSI to one via PCA. The spatial context information is fully utilized in the operation process of the guided filter. In SSBLS, the spatial information is used in the first and third steps. In the second step, BLS uses the spectral-spatial joint features to classify. At the same time, in the third step, the first principal component of spectral information is used to obtain the grey-scale image. Therefore, in the proposed method, the full use of spectral-spatial joint features contributes to better classification performance. The major contribution of our work can be summarized as follows:

(1) We found the organic combination of the Gaussian filter and BLS could enhance the classification accuracy. The Gaussian filter captures the inherent spectral information of each pixel based on HSI spatial information. BLS extracts the sparse and compact features using the random weights fine-turned by the sparse auto encoder in the process of mapping feature. Sparse features can represent the low-level structures such as edges and high-level structures such as local curvatures, shapes [57], these contribute to the improvement of classification accuracy. The inherent spectral features are input to BLS for training and prediction, thereby improving the classification accuracy of the proposed method. Experimental data supports this conclusion.

(2) We take full advantage of spectral-spatial features in SSBLS. The Gaussian filter firstly smooths each spectral band based on spatial information of HSI to achieve the first fusion of spectral-spatial information. The guided filter corrects the results of BLS classification based on the spatial context information again. The grey-scale guidance image of the guided filter is obtained via the first PCA from the original HSI. These three operations sufficiently join spectral information and spatial information together, which is useful to improve the accuracy of SSBLS.

(3) SSBLS utilizes the guided filter to rectify the misclassified hyperspectral pixels based on the spatial contexture information for obtaining the correct classification labels, thereby improving the overall accuracy of SSBLS. The experimental results can also support this point.

The rest of this paper is organized as follows. Section 2 describes the proposed method in detail. Section 3 presents the experiments and analysis. The discussion of the proposed method is in Section 4. Section 5 is the summary.

## 2. Proposed Method of Spectral-Spatial Joint Classification of HSI Based on Broad Learning System

The flowchart of SSBLS proposed in this paper is shown in Figure 1, which mainly consists of three steps: (1) After inputting the original HSI data, the Gaussian filter with an appropriate-sized window is performed to extract the inherent spectral features of samples based on the spatial information. (2) The test samples labels are got using the optimal BLS classification model trained with pixel vectors smoothed by the Gaussian filter. The initial probability maps are constructed according to the results of BLS classification. (3) To improve the classification accuracy of HSI, the guided filter is adopted to correct the initial probability maps based on the spatial context information of HSI under the guiding of the grey-scale guidance image. The guidance image is obtained via the first PCA.

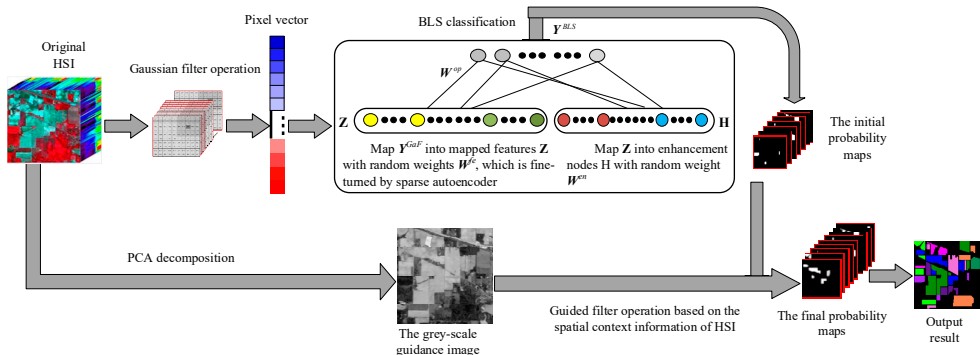

**Figure 1.** The flowchart of hyperspectral image (HSI) classification via the spectral-spatial joint classification broad learning system (SSBLS).

### 2.1. Spectral Feature Extraction of HSI Based on Gaussian Filter

The first step of the proposed method is that the 2-dimensional (2-D) Gaussian filter smooths spectral features on each band based on the spatial information of HSI. The Gaussian filter is one of the most widely used and effective window-based filtering methods. It is usually used as a low-pass filter to suppress the high-frequency noise, and it can repair the detected missing regions [58]. When the Gaussian filter is capturing the spectral features of HSI, the weight of each hyperspectral pixel in the Gaussian filter window decays exponentially according to the distance from the center pixel. The closer the distance of the neighboring pixel from the center pixel is, the greater the weight is, and the farther the distance is, the smaller the weight is. The weight of each pixel in the Gaussian filter window is determined by the following 2-D Gaussian function

$$G(x, y) = \frac{1}{2\pi\sigma^2} e^{-(x^2 + y^2)/2\sigma^2} \tag{1}$$

where $x$ and $y$ are the coordinates of the pixels in the Gaussian filter window on each band of HSI. The coordinate of the center pixel of the window is $(0, 0)$. $\sigma$, is the standard deviation of the Gaussian filter. It is used to control the degree of blurring spectral information. That is to say, the greater the value of $\sigma$ is, the smoother the blurred spectral features are. The Gaussian function [59] has the characteristic of being separable, so that a larger-sized Gaussian filter can be effectively realized. The 2-D Gaussian function convolution can be performed in two steps. First, the spectral image on each band of HSI is convolved with the 1-D Gaussian function, and then, the convolution result is convolved using the same 1-D Gaussian function in the way of rotating 90 degrees to the left. Therefore, the calculation of 2-D Gaussian filtering increases linearly with the size of the filter window instead of increasing squarely.

The original HSI data with $n$ samples are denoted as $X = (x_1, x_2, x_3, \cdots, x_n)$, which belongs to the $m$-D space. $Y^{GaF} = (y_1, y_2, y_3, \cdots y_n) \in R^m$ is gotten from $X$ blurred by the Gaussian filter. Here, $m$ is the number of HSI band. The superscript "*GaF*" represents the

Gaussian filter. The "$O_{GaF}$" stands for the Gaussian filtering operation. The spectral feature extraction of HSI based on the Gaussian filter can be represented as Equation (2).

$$Y^{GaF} = O_{GaF}(X) \tag{2}$$

### 2.2. HSI Classification Based on the Combination of Gaussian Filter and BLS

Chen and Liu put forward a BLS based on the rapid and dynamic learning features of the functional-link network [60–62]. BLS is built as a flat network, in which the input data first are mapped into mapped feature nodes, then all mapped feature nodes are mapped into enhancement nodes for expansion. The BLS network expands through both mapped feature nodes and enhancement nodes. Moreover, through rigorous mathematical methods, Igelnik and Pao [63] have proven that enhancement nodes contribute to the improvement of classification accuracy. BLS is built on the basis of the traditional random vector functional-link neural network (RVFLNN) [64]. However, unlike the traditional RVFLNN, in which the enhancement nodes are constructed though using a linear combination of the input nodes and then applying a nonlinear activation function to them. BLS first maps the inputs to construct a set of mapped feature nodes via some mapping functions and then maps all mapped feature nodes into enhancement nodes through other activation functions.

The second step of the proposed method is to input HSI pixel vectors smoothed by the Gaussian filter to train the BLS classification model. Then the test sample's labels are calculated by the optimal BLS classification model for constructing the initial probability maps. The notation in Table 1 will be used to present the described HSI classification procedure. The HSI samples smoothed by the Gaussian filter are split into a training set and test set. The training pixel vectors are mapped into mapped feature nodes applying the random weight matrix. In addition, the sparse auto encoder is used to fine-tune the random weight matrix. Then, the mapped feature nodes are mapped into enhancement nodes using other random weights. The optimal connection weights from all mapped feature nodes and enhancement nodes to the output are gained through the normalized optimization method of solving L2-norm by ridge regression approximation in order to obtain the optimal BLS model. The test sample labels are predicted by the optimal model to construct the initial probability maps.

**Table 1.** The meaning of notations in BLS classification procedure.

| Notation | Meaning |
| --- | --- |
| $Y^{GaF}$ | The HSI data smoothed by Gaussian filter |
| $Z_i$ | The $i^{th}$ mapped features with $e$ nodes |
| $\phi_i$ | The $i^{th}$ mapping function for feature mapping |
| $W_i^{fe}$ | The $i^{th}$ random weight matrix for feature mapping |
| $\beta_i^{fe}$ | The $i^{th}$ random bias for feature mapping |
| $Z^i$ | The concatenation of all the first $i$ groups of mapping features, $i = 1, \cdots, d$ |
| $Z^d$ | all mapped feature nodes |
| $H_l$ | The $l^{th}$ group of enhancement nodes |
| $\xi_l$ | The $l^{th}$ function for computing the $l^{th}$ group of enhancement nodes |
| $W_l^{en}$ | The $l^{th}$ random weight matrix for computing the $l^{th}$ group of enhancement nodes |
| $\beta_l^{en}$ | The $l^{th}$ random bias for computing the $l^{th}$ group of enhancement nodes |
| $H^l$ | The concatenation of all the first $l$ groups of enhancement nodes |
| $W^{op}$ | The connecting weight matrix from all mapped feature nodes and enhancement nodes to the output |
| $Y^{BLS}$ | The output of BLS |

First, the HSI data smoothed by the Gaussian filter, $\boldsymbol{Y}^{GaF}$ with $n$ samples and $m$ dimensions, is mapped into mapped feature nodes. That is to say, $\boldsymbol{Y}^{GaF} \in R^{n \times m}$. $\boldsymbol{Y}^{BLS} \in R^{n \times C}$ is the result of BLS classification, where $C$ is the quantity of sample types. There are $d$ feature mappings, and each mapping has $e$ nodes, can be represented as in Equation (3) [19]

$$\boldsymbol{Z}_i = \phi_i \left( \boldsymbol{Y}^{GaF} \boldsymbol{W}_i^{fe} + \boldsymbol{\beta}_i^{fe} \right), i = 1, \cdots, d \tag{3}$$

where $\phi_i$ is the mapping function, $\boldsymbol{Z}_i$ is the $i^{th}$ mapped features, $\boldsymbol{W}_i^{fe}$ is a random weight matrix, which has an appropriate dimension, $\boldsymbol{\beta}_i^{fe}$, which is randomly generated, is the bias, "$fe$" represents the mapped feature operation. $\boldsymbol{Z}^i = [\boldsymbol{Z}_1, \cdots, \boldsymbol{Z}_i]$ is the concatenation of all the first $i$ groups of mapping features. Furthermore, $\phi_i$ and $\phi_k$ can be different functions when $i \neq k$. $\boldsymbol{Z}^d = [\boldsymbol{Z}_1, \boldsymbol{Z}_2, \cdots, \boldsymbol{Z}_d]$ represents all mapped feature nodes. In order to capture the sparse and compact features, we make use of the sparse auto encoder to fine-tune the initial $\boldsymbol{W}_i^{fe}$ [19].

Then, Equation (4) is utilized to compute enhancement nodes from mapped feature nodes

$$\boldsymbol{H}_l = \xi_l \left( \boldsymbol{Z}^d \boldsymbol{W}_l^{en} + \boldsymbol{\beta}_l^{en} \right) \tag{4}$$

where $\xi_l$ is an activation function, furthermore, when $l \neq k$, $\xi_l$ and $\xi_k$ can be different functions. $\boldsymbol{H}_l$ is the $l^{th}$ group of enhancement nodes, $\boldsymbol{W}_l^{en}$, which has appropriate dimensions, is a randomly generated weight matrix. $\boldsymbol{\beta}_l^{en}$, which is randomly generated, is the bias. The process of mapping enhancement nodes is used the "$en$" to represent. $\boldsymbol{H}^l = [\boldsymbol{H}_1, \cdots, \boldsymbol{H}_l]$ is the concatenation of all the first $l$ groups of enhancement nodes [19].

Combined with Equation (4), the output result of BLS can be expressed by Equation (5)

$$\begin{aligned} \boldsymbol{Y}^{BLS} &= \left[ \boldsymbol{Z}_1, \cdots, \boldsymbol{Z}_d \middle| \xi_1 \left( \boldsymbol{Z}^d \boldsymbol{W}_1^{en} + \boldsymbol{\beta}_1^{en} \right), \cdots, \xi_l \left( \boldsymbol{Z}^d \boldsymbol{W}_l^{en} + \boldsymbol{\beta}_l^{en} \right) \right] \boldsymbol{W}^{op} \\ &= \left[ \boldsymbol{Z}_1, \cdots, \boldsymbol{Z}_d \middle| \boldsymbol{H}_1, \cdots, \boldsymbol{H}_l \right] \boldsymbol{W}^{op} \\ &= \left[ \boldsymbol{Z}^d \middle| \boldsymbol{H}^l \right] \boldsymbol{W}^{op} \end{aligned} \tag{5}$$

where $\boldsymbol{W}^{op}$ is the connecting weight matrix from all mapped feature nodes and all enhancement nodes to the output of the BLS. The superscript "$op$" represents the optimal weight [19]. The optimal connecting weight matrix can be obtained using the L2-norm regularized least square problem as shown in Equation (6)

$$\underset{\boldsymbol{W}^{op}}{\arg \min} : \left\| \left[ \boldsymbol{Z}^d \middle| \boldsymbol{H}^l \right] \boldsymbol{W}^{op} - \boldsymbol{Y}^{GaF} \right\|_2^2 + \lambda \left\| \boldsymbol{W}^{op} \right\|_2^2 \tag{6}$$

where $\lambda$ is applied to further restrict the squared of L2-norm of $\boldsymbol{W}^{op}$. $\| \cdot \|_2$ represents the L2-norm, and $\| \cdot \|_2^2$ stands for the square of L2-norm. Equation (7) is obtained by the ridge regression approximation [19].

$$\boldsymbol{W}^{op} = \left( \lambda \boldsymbol{I} + \left[ \boldsymbol{Z}^d \middle| \boldsymbol{H}^l \right] \left[ \boldsymbol{Z}^d \middle| \boldsymbol{H}^l \right]^T \right)^{-1} \left[ \boldsymbol{Z}^d \middle| \boldsymbol{H}^l \right]^T \boldsymbol{Y}^{GaF} \tag{7}$$

When $\lambda \to 0$, Equation (7) can be converted into solving the least square problem. When $\lambda \to \infty$, the result of Equation (7) is finite and turns to zero. So, set $\lambda \to 0$, and add a positive number on the diagonal of $\left[ \boldsymbol{Z}^d \middle| \boldsymbol{H}^l \right]^T \left[ \boldsymbol{Z}^d \middle| \boldsymbol{H}^l \right]$ or $\left[ \boldsymbol{Z}^d \middle| \boldsymbol{H}^l \right] \left[ \boldsymbol{Z}^d \middle| \boldsymbol{H}^l \right]^T$ to get the approximate Moore-Penrose generalized inverse [19]. Consequently, we have Equation (8).

$$\left[ \boldsymbol{Z}^d \middle| \boldsymbol{H}^l \right]^+ = \lim_{\lambda \to 0} \left( \left( \lambda \boldsymbol{I} + \left[ \boldsymbol{Z}^d \middle| \boldsymbol{H}^l \right] \left[ \boldsymbol{Z}^d \middle| \boldsymbol{H}^l \right]^T \right)^{-1} \right) \left[ \boldsymbol{Z}^d \middle| \boldsymbol{H}^l \right]^T \tag{8}$$

So we have:

$$W^{op} = \left[ \mathbf{Z}^d \middle| \mathbf{H}^l \right]^+ Y^{GaF} \tag{9}$$

Finally, the output of BLS is:

$$Y^{BLS} = \left[ \mathbf{Z}^d \middle| \mathbf{H}^l \right] \left[ \mathbf{Z}^d \middle| \mathbf{H}^l \right]^+ Y^{GaF} \tag{10}$$

After inputting the spectral features smoothed by the Gaussian filter into BLS, the initial result of classification is $Y^{BLS} = (y_1^{BLS}, y_2^{BLS}, \cdots y_n^{BLS})$. The probability maps of this results are expressed as $\mathbf{P} = (\mathbf{p}_1, \cdots, \mathbf{p}_C)$, here $\mathbf{p}_c$ is the probability map that all pixels belong to the $c$ class. $p_{i,c} \in \mathbf{p}_c, i = 1, 2, \cdots, n$ is the probability that the pixel $i$ belongs to $c$ ($c = 1, 2, \cdots C$). Specifically, as followed Equation (11).

$$p_{i,c} = \begin{cases} 1 & \text{if } y_i^{BLS} = c \\ 0 & \text{otherwize} \end{cases} \tag{11}$$

### 2.3. Correction to the Results of BLS Classification Based on Guided Filter

In the third step of the proposed method, the guided filter is performed to correct each probability map $\mathbf{p}_c$ with the guidance of the grey-scale guidance image $V$, and get the output $\mathbf{q}_c$ ($c = 1, 2, \cdots C$). $V$ is obtained by the first PCA method from the original HSI. The output of the guided filter [38] is the local linear transformation of the guidance image and has a good edge-preserving characteristic. At the same time, the output image will become more structured and non-smooth than the input image under the guidance of the guidance image. For grey-scale and high-dimensional images, the guided filter essentially has the characteristic of low time complexity, regardless of the kernel size and the intensity range. In this step, the filtering output is $\mathbf{Q} = (\mathbf{q}_1, \mathbf{q}_2, \cdots \mathbf{q}_C)$ ($c = 1, 2, \cdots C$). Here, $\mathbf{q}_c$ is the probability map that all pixels belong to the $c$ class. $q_{i,c} \in \mathbf{q}_c, i = 1, 2, \cdots, n$, which is the probability that the pixel $i$ belongs to $c$ ($c = 1, 2, \cdots C$), can be expressed as a linear transformation of the guidance image in a window $\omega_k$ centered at the pixel $k$, as shown in Equation (12).

$$q_{i,c} = a_k V_i + b_k, \forall i \in \omega_k \tag{12}$$

$(a_k, b_k)$ are some assumed linear coefficients to be restricted in $\omega_k$. $\omega_k$ is a window, the radius of which is $r$. This local linear model guarantees that $\mathbf{q}_c$ has an edge only if $V$ has an edge, because $\nabla \mathbf{q}_c \approx a \nabla V$. The cost function in the window $\omega_k$ is minimized as shown in Equation (13), which can not only realize the linear model of Equation (12), but also minimize the difference between $\mathbf{q}_c$ and $V$.

$$E(a_k, b_k) = \sum_{i \in \omega_k} \left( (a_k V_i + b_k - p_{i,c})^2 + \varepsilon a_k^2 \right) \tag{13}$$

$\varepsilon$, which defines the degree of the guided filter blurring, is used to regularize the parameter penalizing large $a_k$. Equation (13) is the linear ridge regression model and is solved by Equation (14).

$$a_k = \frac{\frac{1}{|\omega|} \sum_{i \in \omega_k} V_i p_{i,c} - \mu_k \overline{p}_{k,c}}{\sigma_k^2 + \varepsilon} \tag{14}$$

$$b_k = \overline{p}_{k,c} - a_k \mu_k \tag{15}$$

Here $\mu_k$ and $\sigma_k^2$ are the mean and variance of the guidance image in $\omega_k$. $|\omega|$ is the quantity of pixels in $\omega_k$. $\overline{p}_{k,c} = \frac{1}{|\omega|} \sum_{i \in \omega_k} p_{i,c}$ is the mean of $\mathbf{p}_c$ in $\omega_k$.

Pixel $i$ is involved in all the overlapping windows, which cover pixel $i$; therefore, the value of $q_{i,c}$ in Equation (12) is different in different windows. $q_{i,c}$ can be acquired by averaging all the possible values which are computed in different windows. So, after

calculating $a_k$ and $b_k$ for all windows $\omega_k$ in $V$, the output is calculated by using Equation (16) as follows:

$$q_{i,c} = \frac{1}{|\omega|} \sum_{k|i \in \omega_k} (a_k V_i + b_k) \tag{16}$$

The window $\omega_k$ is symmetrical, so $\sum_{k|i \in \omega_k} a_k = \sum_{k \in \omega_i} a_k$, Equation (16) can be expressed by Equation (17)

$$q_{i,c} = \overline{a}_i V + \overline{b}_i \tag{17}$$

where $\overline{a}_i = \frac{1}{|\omega|}\sum_{k \in \omega_i} a_k$ and $\overline{b}_i = \frac{1}{|\omega|}\sum_{k \in \omega_i} b_k$ are the mean coefficients of all windows covering pixel $i$.

In fact, $a_k$ in Equation (14) can be rewritten as a weighted sum of input image $\mathbf{p}_c$: $a_k = \sum_j A_{k,j}(V)p_{j,c}$, $A_{i,j}$ is the weight that only depends on the guiding image $V$. Similarly, $b_k = \sum_j B_{k,j}(V)p_{j,c}$. The kernel weight is explicitly expressed by:

$$\theta_{i,j}(V) = \frac{1}{|\omega|^2} \sum_{k \in \omega_i, k \in \omega_j} \left( 1 + \frac{(V_i - \mu_k)(V_j - \mu_k)}{\sigma_k^2 + \varepsilon} \right) \tag{18}$$

So, Equation (17) can be changed to Equation (19).

$$q_{i,c} = \sum_j \theta_{i,j}(V)p_{j,c} \tag{19}$$

After the initial probability maps are corrected by the guided filter, the probability of the pixel $i$ has $C$ values, that is to say, $q_{i,c}, c = 1, 2, \cdots, C$. We take the subscript of the highest probability among the $C$ probabilities as the label of the pixel $i$, namely:

$$y_i^{GuF} = \arg \max_c q_{i,c}, c = 1, 2, \cdots, C \tag{20}$$

After the guided filter corrects the initial probability maps, the labels of all labeled samples of HSI are $Y^{GuF} = \left(y_1^{GuF}, y_2^{GuF}, \cdots, y_n^{GuF}\right)$. The superscript "*GuF*" represents the guided filtering operation.

In summary, the algorithmic steps of HSI classification based on SSBLS are summarized in Algorithm 1.

---

**Algorithm 1. Algorithmic details of SSBLS**

---

1. ***Input:*** Original HSI Dataset, $X$; $S$ is the size of the Gaussian filter window; $\sigma$ is the standard deviation of the Gaussian function; $N$ is the number of training samples; $M$ is the number of mapped feature windows; $F$ is the number of mapped feature nodes per window; $E$ is the number of enhancement nodes; $r$ represents the radius of the guided filter window $\omega_k$; $\varepsilon$ is the penalty parameter of the guided filter.
2. Select the optimal parameters $S$ and $\sigma$, perform Gaussian filter to smooth each spectral band of original HSI data $X$ and get $Y^{GaF}$.
3. Randomly select $N$ samples from the labeled samples of each class of HSI as the training set $Y_{train}^{GaF}$, and the remaining labeled samples are the test sample set $Y_{test}^{GaF}$.
4. Randomly produce $W_i^{fe}$ with $F$ columns and $\boldsymbol{\beta}_i^{fe}$, fine-tune $W_i^{fe}$ using the sparse auto encoder, and get $Z_{train,i}$ according to the $Y_{train}^{GaF}$, $W_i^{fe}$, $\boldsymbol{\beta}_i^{fe}$ and Equation (3), where $i = 1, 2, \cdots, M, F = e$.
5. Set random weights $W^{fe} = \left[W_1^{fe}, \cdots, W_M^{fe}\right]$, random bias $\boldsymbol{\beta}^{fe} = \left[\boldsymbol{\beta}_1^{fe}, \cdots, \boldsymbol{\beta}_M^{fe}\right]$, and the feature mapping group $\mathbf{Z}_{train}^d = \left[\mathbf{Z}_{train,1}, \cdots, \mathbf{Z}_{train,M}\right]$, where $M = d$.
6. Randomly produce $W^{en}$ with $F \times M$ rows and $E$ columns and $\boldsymbol{\beta}^{en}$, map $\mathbf{Z}_{train}^d$ into $\mathbf{H}_{train}^l$ using Equation (4).
7. Obtain the optimal connecting weights $W^{op}$ according to Equations (8) and (9).

---

8. Map $\boldsymbol{Y}_{test}^{GaF}$ into $\boldsymbol{Z}_{test}^{d}$ with $\boldsymbol{W}^{fe}$, $\boldsymbol{\beta}^{fe}$ using Equation (3), and map $\boldsymbol{Z}_{test}^{d}$ into $\boldsymbol{H}_{test}^{l}$ by Equation (4) with $\boldsymbol{W}^{en}$ and $\boldsymbol{\beta}^{en}$, compute the test samples labels $\boldsymbol{Y}_{test}^{BLS}$ with $\boldsymbol{W}^{op}$ and Equation (5).

9. Construct the initial probability maps of the entire HSI based on the labels of training samples and test samples, namely $\mathbf{P} = (\mathbf{p}_1, \mathbf{p}_2, \cdots, \mathbf{p}_C)$.

10. Based on the original HSI, the grey-level guidance map $\boldsymbol{V}$ is generated by the first PCA method. According to Equations (18) and (19), and the optimal parameters $r$ and $\varepsilon$, correct each initial probability map $\mathbf{p}_c$ respectively, then get the final probability graphs $\mathbf{q} = (\mathbf{q}_1, \mathbf{q}_2, \cdots \mathbf{q}_C)$, where $c = 1, 2, \cdots, C$.

11. According to Equations (20), based on the maximum probability principle, get $\boldsymbol{Y}^{GuF}$, the classification results of all samples, get the test samples labels after re-moving the training samples.

12. *Output:* the test sample's labels.

## 3. Experiment Results

We assess the proposed SSBLS through a lot of experiments. All experiments are performed in MATLAB R2014a using a computer with 2.90 GHz Intel Core i7-7500U central processing unit (CPU) and 32 GB memory and Windows 10.

### 3.1. Hyperspectral Image Dataset

The performance of SSBLS method and other comparison methods are evaluated on the three public hyperspectral datasets, which are the Indian Pines, Salinas, and Pavia University datasets (The three datasets are available at http://www.ehu.eus/ccwintco/index.php?title=Hyperspectral_Remote_Sensing_Scenes accessed on 4 November 2018).

The Indian Pines dataset was acquired by the Airborne Visible Infrared Imaging Spectrometer (AVIRIS) sensor when it was flying over North-west Indiana Indian Pine test site. This scene has 21,025 pixels and 200 bands. The wavelength of bands is from 0.4 to 2.5 µm. Two-thirds agriculture and one-third forests or other perennial natural vegetation constitute this image. There are two main two-lane highways, a railway line, some low-density housing, other built structures, and pathways in this image. It has 16 types of things. In our experiments, we selected the nine categories samples with a quantity greater than 400. The original hyperspectral image and ground truth are given in Figure 2.

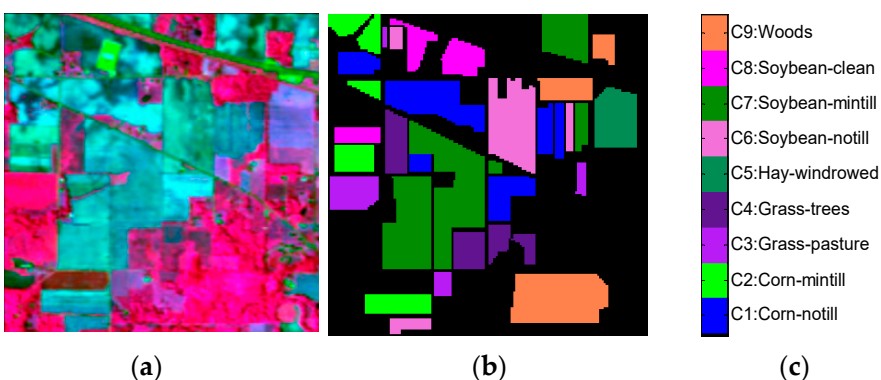

(a)  (b)  (c)

**Figure 2.** Indian Pines dataset with (**a**) original hyperspectral image, (**b**) ground truth, and (**c**) category names with labeled samples.

The Salinas scene was obtained by a 224-band AVIRIS sensor, capturing over the Salinas Valley, California, USA, with a high spatial resolution of 3.7 m. The HSI dataset has $512 \times 217$ pixels with 204 bands after the 20 water absorption bands were discarded. We made use of 16 classes samples in the scene. The original hyperspectral image and ground truth are given in Figure 3.

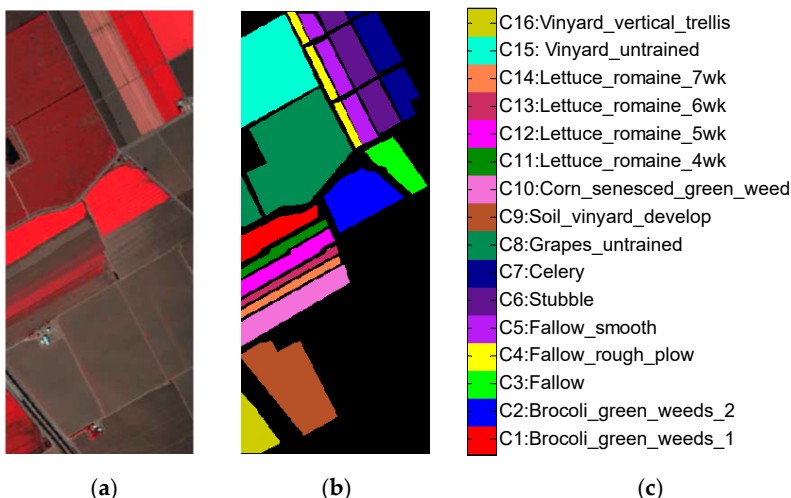

**Figure 3.** Salinas dataset with (**a**) original hyperspectral image, (**b**) ground truth, and (**c**) category names with labeled samples.

The Pavia University dataset was collected by the Reflective Optics System Imaging Spectrometer (ROSIS) sensor over Pavia in northern Italy. The image has $610 \times 340$ pixels with 103 bands. Some pixels containing nothing in the image were removed. There were nine different sample categories used in our experiments. Figure 4 is the original hyperspectral image, category names with labeled samples, and ground truth.

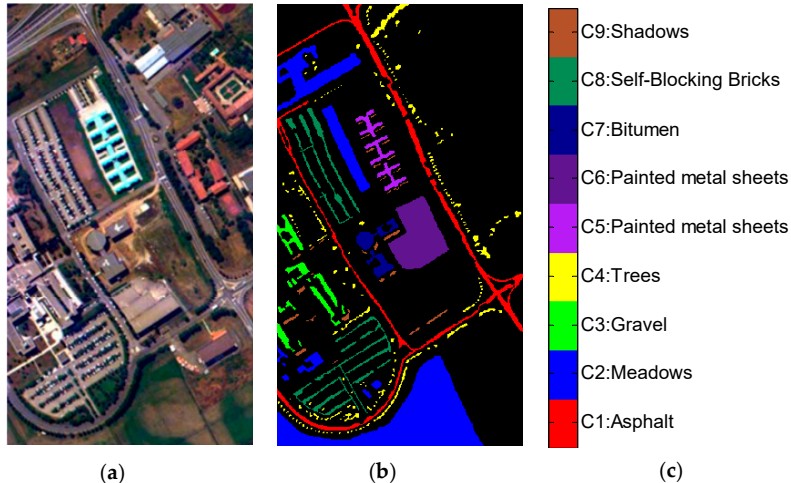

**Figure 4.** Pavia University dataset with (**a**) original hyperspectral image, (**b**) ground truth, and (**c**) category names with labeled samples.

### 3.2. Parameters Analysis

After analyzing SSBLS, it was found that the adjustable parameters are the size of the Gaussian filter window ($S$), the standard deviation of Gaussian function ($\sigma$), the number of mapped feature windows in BLS ($M$), the number of mapped feature nodes per window in BLS ($F$), the number of enhancement nodes ($E$), the radius of the guided filter window $\omega_k$ ($r$), and the penalty parameter of the guided filter ($\varepsilon$). The above parameters are analyzed with overall accuracy (OA) to evaluate the performance of SSBLS.

### 3.2.1. Influence of Parameter $S$ and $\sigma$ on OA

In this section, the influence of $S$ and $\sigma$ on OA was analyzed in three datasets. $S$ and $\sigma$ are took different values, and other parameters are fixed values, namely, $M = 20, F = 40, E = 1000, r = 2, \varepsilon = 0.001$. $S$ was chosen from $[2, 4, 6, \cdots, 30]$, and the value range of $\sigma$ was

chosen from $[1, 2, 3, \cdots, 15]$ in the Indian Pines and Salinas datasets. $S$ and $\sigma$ were chosen from $[1, 3, 5, \cdots, 29]$ and $[1, 2, 3, \cdots, 15]$, respectively, in the Pavia University dataset. The mean OAs of ten experiments are shown in Figure 5. It can be seen from this figure that as the $S$ and $\sigma$ increased, the OAs gradually increased, and gradually decreased after reaching the peak. If $S$ is too small, the larger-sized target will divide into multiple parts distributing in the diverse Gaussian filter windows. If $S$ is too large, the window will contain multiple small-sized targets. Both will cause misclassification. When $\sigma$ is too small, the weights change drastically from the center to the boundary. When $\sigma$ gradually becomes larger, the weights change smoothly from the center to the boundary, and the weights of pixels in the window are relatively well-distributed, which is close to the mean filter. Therefore, for different HSI datasets, the optimal values of $S$ and $\sigma$ were not identical. in the Indian Pines dataset, when $S = 18, \sigma = 7$, the OA is the largest. So $S$ and $\sigma$ were 18 and 7 respectively in the subsequent experiments. In the Salinas dataset, when $S = 24, \sigma = 7$, the performance of SSBLS was the best. Therefore, $S$ and $\sigma$ were taken as 24 and 7 in the later experiments severally. Similarly, the best values of $S$ and $\sigma$ were 21 and 4 respectively in the Pavia University dataset.

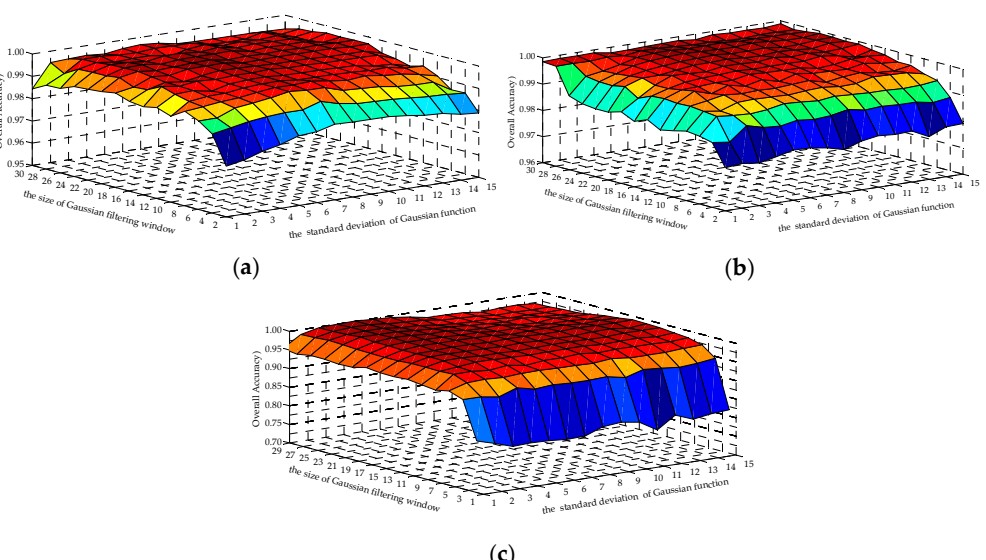

**Figure 5.** The relationship of the Gaussian filter window ($S$), standard deviation of the Gaussian function ($\sigma$ ), and overall accuracy (OA) in the three datasets. (**a**) Indian Pines; (**b**) Salinas; (**c**) Pavia University.

### 3.2.2. Influence of Parameter $M$ and $F$ on OA

The experiments are carried out on the three datasets. The values of $S$ and $\sigma$ were the optimal values obtained from the above analysis, and $E = 1000, r = 2, \varepsilon = 0.001$. In the Indian Pines and Pavia University datasets, $M$ and $F$ were chosen from $[2, 4, 6, \cdots, 20]$ and $[2, 6, 10, \cdots, 38]$. The values range of $M$ and $F$ were $[2, 4, 6, \cdots, 20]$ and $[4, 8, 12, \cdots, 40]$ in the Salinas dataset. As shown in Figure 6, we can see that as $M$ and $F$ were becoming larger, the OAs of SSBLS gradually grew. When $M$ and $F$ were too small, the lesser feature information was extracted and the lower the mean OA of ten experiments was. When $M$ and $F$ were too large, although the performance of SSBLS was improved, the computation and the consumed time also rose. Therefore, in the subsequent experiments, the best values of $M$ and $F$ were 6 and 34 respectively in the Indian Pines dataset, 12 and 36 in the Salinas dataset, and 8 and 26 in the Pavia University dataset.

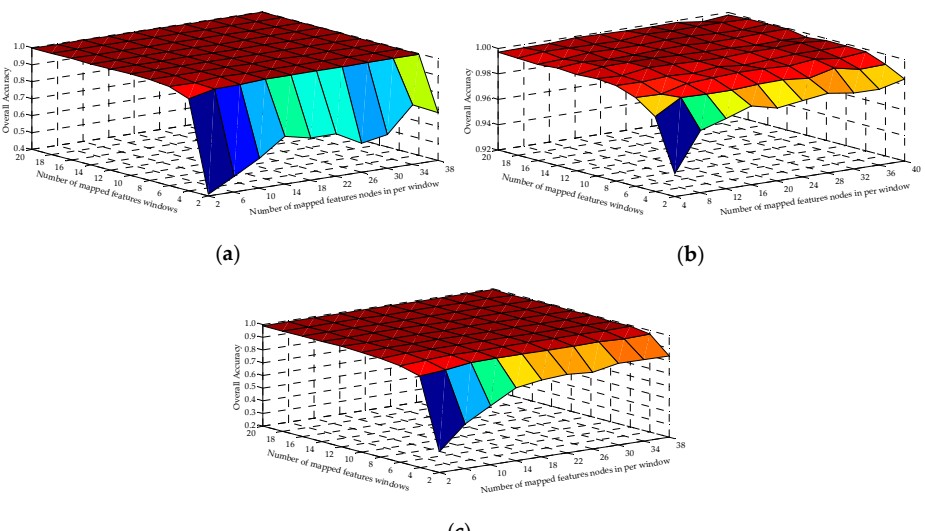

**Figure 6.** The relationship of the number of mapped feature windows in BLS (*M*), the number of mapped feature nodes per window in BLS (*F*), and the OA in the three datasets. (**a**) Indian Pines; (**b**) Salinas; (**c**) Pavia University.

### 3.2.3. Influence of Parameter *E* on OA

In the three datasets, *S*, *σ*, *M* and *F* were the optimal values obtained from the above experiments, *r* and *ε* were 2 and $10^{-3}$, respectively. *E* was chosen from $[500, 550, 600, \cdots, 1200]$ in the Indian Pines dataset. The range of *E* was $[50, 100, 150, \cdots, 800]$ in the Salinas and Pavia University datasets. In the three datasets, the average OAs of ten experiments had an upward trend with the increase of *E* as shown in Figure 7. As *E* gradually grew, the features extracted by BLS also increased, at the same time, the computation and consumed time also grew. Therefore, the numbers of enhanced nodes were 1050 in the Indian Pines dataset, and 700 in both the Salinas and Pavia University datasets.

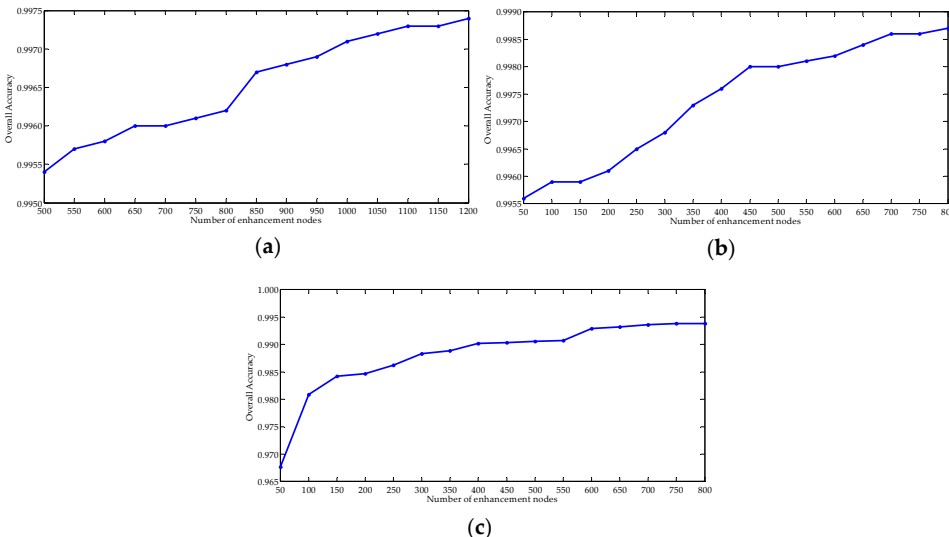

**Figure 7.** The relationship of the number of enchancement nodes (*E*) and OA in the three datasets. (**a**) Indian Pines; (**b**) Salinas; (**c**) Pavia University.

### 3.2.4. Influence of Parameter *r* on OA

The experiments were carried out on the three datasets. The values of *S*, *σ*, *M*, *F* and *E* were the optimal values analyzed previously, *ε* is $10^{-3}$, and *r* is chosen from $[1, 2, 3, \cdots 9]$. Figure 8 indicates that as *r* grew, the average OAs of ten experiments first increased, and

then decreased. In the Indian Pines dataset, when $r = 3$, the mean OA was the largest, so $r$ is 3. In the Salinas dataset, when $r = 5$, the performance of SSBLS was the best, so the value of $r$ was 5. On the Pavia University dataset, while $r = 3$, the average OA was the greatest, so $r$ was 3.

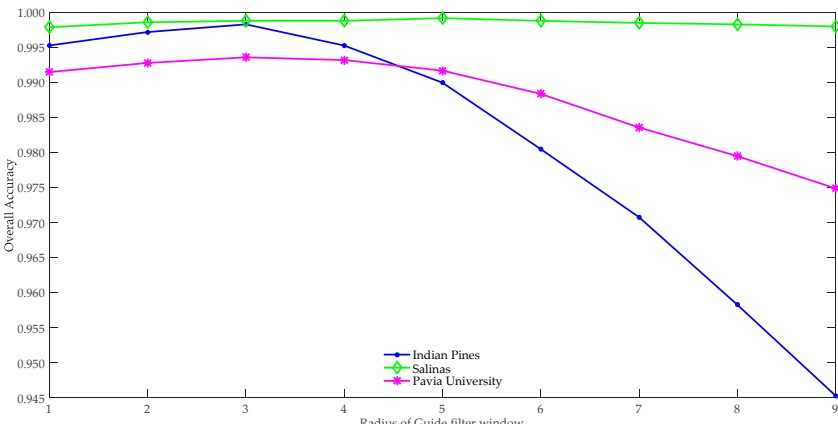

**Figure 8.** The relationship of OA and $r$ in the three datasets.

### 3.2.5. Influence of Parameter $\varepsilon$ on OA

In the three datasets, $S$, $\sigma$, $M$, $F$, $E$ and $r$ were the optimal values obtained in the above experiments. The value range of $\varepsilon$ was $[10^{-7}, 10^{-6}, 10^{-5}, \cdots, 1]$. In the Indian Pines and Salinas datasets, as $\varepsilon$ increased, the mean OAs first increased and then decreased, as shown in Figure 9. In the Indian Pines dataset, when $\varepsilon = 10^{-3}$, the average OA was the largest, so $\varepsilon$ was $10^{-3}$ in the subsequent compared experiments. On the Salinas dataset, while $\varepsilon = 10^{-1}$, the performance of SSBLS was the best, so the optimal value of $\varepsilon$ was $10^{-1}$. In the Pavia University dataset, as $\varepsilon = 10^{-7}$, the classification effect was the best, then the best value of $\varepsilon$ was $10^{-7}$.

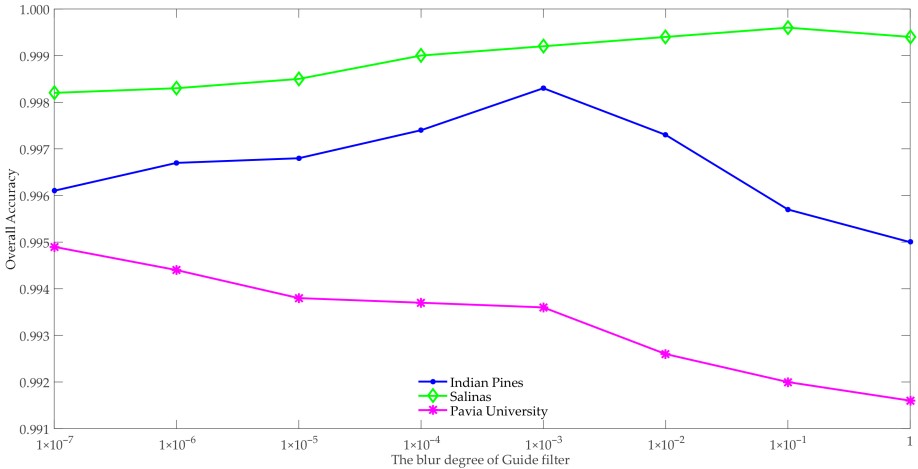

**Figure 9.** The relationship of OA and $\varepsilon$ in the three datasets.

### 3.3. Ablation Studies on SSBLS

We have conducted several ablation experiments to investigate the behavior of SSBLS on the three datasets. In these ablation experiments, we randomly took 200 labeled samples as training samples and the remaining labeled samples as test samples from each class sample. We utilized OA, average accuracy (AA), kappa coefficient (Kappa) to measure the performance of different methods as shown Table 2, and the highest values of them are shown in bold.

First, we only used BLS to classify the original hyperspectral data. On the Salinas dataset, the effect was good; the OA reached 91.98%. However, the results were unsatisfactory when using the Indian Pines and Pavia University datasets.

Second, we disentangle the Gaussian filter influence on the classification results. We used the Gaussian filter to smooth the original HSI, and then used BLS to classify, namely the method of BLS based on the Gaussian filter (GBLS). In Indian Pines dataset, the OA was about 20% higher than these of BLS, about 7% higher than that of BLS in the Salinas dataset, and about 17% higher in the Pavia University dataset. These show that the Gaussian filter can help to improve the classification accuracy.

Next, we used BLS to classify the original hyperspectral data and then applied the guided filter to rectify the misclassified pixels of BLS. The results in terms of OA, AA, and Kappa were also better than those of BLS. This shows that guided filter also plays a certain role in improving classification performance.

Finally, we used the proposed method in the paper for HSI classification. This method first uses the Gaussian filter to smooth the original spectral features based on the spatial information of HSI. After using BLS classification, it finally applies the guided filter to correct the pixels that are misclassified by BLS. The results are the best in the four methods. This shows that both Gaussian filter and guided filter contribute to the improvement of classification performance.

From the above analysis, we know that the combination of the Gaussian filtering and BLS has a great effect on the improvement of classification performance, especially on Indian Pines and Pavia University datasets. Although the classification accuracy after BLS classification based on the Gaussian filter (GBLS) was relatively high, the classification accuracy was still improved after adding the guided filter to GBLS. It indicates that the guided filter can also help improve the classification accuracy.

### 3.4. Experimental Comparison

In order to prove the advantages of SSBLS on the three real datasets, we compare SSBLS with SVM [65], HiFi-We [42], SSG [66], spectral-spatial hyperspectral image classification with edge-preserving filtering (EPF) [41], support vector machine based on the Gaussian filter (GSVM), feature extraction of hyperspectral images with image fusion and recursive filtering (IFRF) [67], LPP_LBP_BLS [19], BLS [50], and GBLS. All methods inputs are the original HSI data. Furthermore, the experimental parameters are the optimal values. In each experiment, the 200 labeled samples are randomly selected from per class sample as the training set, and the rest labeled samples as the test samples set. We get the individual classification accuracy (ICA), OA, AA, Kappa, overall consumed time (*t*), and test time (*tt*). All results are the mean values of ten experiments as shown in Tables 3–5, and the highest values of them are shown in bold.

**Table 2.** The results of ablation experiments on the three datasets in term of overall accuracy, OA (%), average accuracy, AA (%), and kappa coefficient, Kappa (%).

| Method | | BLS | GaussianF+BLS | BLS+GuidedF | SSBLS |
|---|---|---|---|---|---|
| Indian Pines | OA | 78.32 | 99.32 | 96.69 | **99.83** |
| | AA | 80.66 | 99.19 | 96.84 | **99.86** |
| | Kappa | 74.29 | 99.18 | 96.04 | **99.80** |
| Salinas | OA | 91.98 | 99.74 | 96.84 | **99.96** |
| | AA | 96.26 | 99.80 | 98.82 | **99.97** |
| | Kappa | 91.04 | 99.71 | 96.46 | **99.95** |
| Pavia University | OA | 70.23 | 99.15 | 85.77 | **99.49** |
| | AA | 70.21 | 98.77 | 81.50 | **99.35** |
| | Kappa | 61.22 | 98.86 | 87.10 | **99.32** |

**Table 3.** Classification results of all comparison methods on the Indian Pines dataset in term of individual classification accuracy, ICA (%), overall accuracy, OA (%), average accuracy, AA (%), kappa coefficient, Kappa (%), overall consumed time, *t* (s), and test time, *tt* (s). SVM: support vector machine; HiFi-We: hierarchical guidance filtering-based ensemble classification for hyperspectral images; EPF: edge-preserving filtering; GSVM: Gaussian support vector machine; IFRF: image fusion and recursive filtering; LLP_LBP_BLS: locality preserving projections local binary pattern broad learning system.

| | Class | SVM | HiFi-We | SSG | EPF | GSVM | IFRF | LPP_LBP_BLS | BLS | GBLS | SSBLS |
|---|---|---|---|---|---|---|---|---|---|---|---|
| | C1 | 77.43 | 85.48 | 53.21 | 94.53 | 90.77 | 97.74 | 99.63 | 72.59 | 98.61 | **99.48** |
| | C2 | 77.75 | 82.32 | 60.81 | 95.47 | 95.89 | 98.18 | 99.68 | 59.56 | 99.72 | **100.00** |
| | C3 | 95.09 | 90.02 | 92.08 | 93.22 | 97.39 | 99.68 | **100.00** | 90.23 | 98.29 | 99.93 |
| | C4 | 98.66 | 96.47 | 97.96 | 96.17 | 99.08 | 99.10 | **100.00** | 97.22 | 97.79 | 99.63 |
| ICA | C5 | 99.86 | 99.75 | 99.21 | **100.00** | 99.96 | **100.00** | **100.00** | 99.85 | 99.72 | **100.00** |
| | C6 | 81.04 | 69.56 | 71.58 | 86.35 | 95.23 | 96.40 | 99.79 | 60.94 | 99.10 | **99.95** |
| | C7 | 64.94 | 92.24 | 55.49 | 97.69 | 95.04 | 99.59 | 99.27 | 82.49 | 99.85 | **99.87** |
| | C8 | 83.56 | 60.79 | 60.28 | 92.90 | 99.49 | 98.74 | **100.00** | 63.52 | 99.85 | 99.90 |
| | C9 | 97.83 | 99.47 | 94.23 | 99.52 | 99.33 | **100.00** | 99.85 | 99.53 | 99.75 | **100.00** |
| OA | | 80.31 | 86.14 | 69.09 | 95.38 | 95.84 | 98.80 | 99.74 | 78.32 | 99.32 | **99.83** |
| AA | | 86.24 | 86.23 | 76.09 | 95.09 | 96.91 | 98.83 | 99.80 | 80.66 | 99.19 | **99.86** |
| Kappa | | 76.79 | 83.61 | 63.80 | 94.48 | 95.00 | 98.56 | 99.64 | 74.29 | 99.18 | **99.80** |
| *t* | | 2.15 | 83.26 | 440.71 | 160.70 | 2.26 | 27.73 | 113.45 | **0.80** | 1.25 | 1.42 |
| *tt* | | 1.47 | 0.64 | 285.99 | 4.48 | 0.87 | 0.35 | 0.48 | **0.31** | **0.31** | 0.45 |

**Table 4.** Classification results of all comparison methods on the Salinas dataset in term of individual classification accuracy, ICA (%), overall accuracy, OA (%), average accuracy, AA (%), kappa coefficient, Kappa (%), overall consumed time, *t* (s), and test time, *tt* (s).

| | Class | SVM | HiFi-We | SSG | EPF | GSVM | IFRF | LPP_LBP_BLS | BLS | GBLS | SSBLS |
|---|---|---|---|---|---|---|---|---|---|---|---|
| | C1 | 99.62 | 99.97 | 98.03 | **100.00** | 99.76 | **100.00** | **100.00** | 99.78 | **100.00** | **100.00** |
| | C2 | 99.74 | 99.25 | 92.31 | 99.92 | 99.74 | **100.00** | **100.00** | 99.91 | **100.00** | **100.00** |
| | C3 | 99.60 | 96.40 | 77.99 | 98.91 | 99.30 | 99.92 | **100.00** | 98.33 | **100.00** | **100.00** |
| | C4 | 99.61 | 97.70 | 99.45 | 98.87 | 97.31 | 98.20 | **100.00** | 98.84 | 98.84 | 99.85 |
| | C5 | 98.54 | 97.37 | 95.28 | 99.76 | 98.50 | **99.98** | 99.40 | 98.87 | 99.71 | 99.90 |
| | C6 | 99.78 | **100.00** | 99.60 | 99.97 | 99.22 | 99.98 | 99.48 | 99.88 | 99.85 | 99.97 |
| | C7 | 99.66 | 99.34 | 98.04 | 99.81 | 99.71 | 99.87 | **100.00** | 99.91 | **100.00** | **100.00** |
| ICA | C8 | 84.47 | 84.44 | 58.46 | 91.46 | 88.38 | 99.73 | 99.73 | 84.95 | **100.00** | **100.00** |
| | C9 | 99.64 | 99.08 | 91.65 | 99.50 | 99.78 | **100.00** | **100.00** | 99.35 | **100.00** | **100.00** |
| | C10 | 95.64 | 90.56 | 75.36 | 96.42 | 99.28 | 99.98 | 99.97 | 97.35 | **100.00** | **100.00** |
| | C11 | 99.33 | 89.32 | 78.55 | 98.84 | **100.00** | 99.08 | 99.88 | 98.10 | 99.99 | **100.00** |
| | C12 | 99.97 | 94.34 | 99.52 | 99.90 | 99.63 | **100.00** | 99.94 | 98.77 | **100.00** | 99.96 |
| | C13 | 99.59 | 96.21 | 97.61 | 99.78 | 99.59 | 99.83 | **100.00** | 99.89 | 99.97 | 99.97 |
| | C14 | 98.21 | 85.68 | 91.84 | 97.57 | 99.83 | 98.92 | **100.00** | 95.28 | 99.87 | **100.00** |
| | C15 | 69.74 | 69.28 | 68.68 | 85.45 | 98.25 | 99.10 | 99.72 | 71.19 | 98.57 | **99.79** |
| | C16 | 98.87 | 97.97 | 88.19 | 99.24 | 99.99 | 99.97 | **100.00** | 99.82 | **100.00** | **100.00** |
| OA | | 91.87 | 90.31 | 81.45 | 95.63 | 96.88 | 99.72 | 99.83 | 91.98 | 99.74 | **99.96** |
| AA | | 96.38 | 93.56 | 88.16 | 97.84 | 98.74 | 99.66 | 99.88 | 96.26 | 99.80 | **99.97** |
| Kappa | | 90.90 | 89.17 | 79.37 | 95.11 | 96.51 | 99.68 | 99.81 | 91.04 | 99.71 | **99.95** |
| *t* | | 9.21 | 167.57 | 1308.90 | 317.40 | 9.53 | 57.13 | 217.34 | **4.11** | 5.06 | 6.10 |
| *tt* | | 7.54 | 3.10 | 136.23 | 16.33 | 7.21 | **1.20** | 4.96 | 2.15 | 2.20 | 3.16 |

**Table 5.** Classification results of all comparison methods on the Pavia University dataset in term of individual classification accuracy, ICA (%), overall accuracy, OA (%), average accuracy, AA (%), kappa coefficient, Kappa (%), overall consumed time, *t* (s), and test time, *tt* (s).

| | Class | SVM | HiFi-We | SSG | EPF | GSVM | IFRF | LPP_LBP_BLS | BLS | GBLS | SSBLS |
|---|---|---|---|---|---|---|---|---|---|---|---|
| | C1 | 95.24 | 93.19 | 64.43 | 99.00 | 93.74 | 97.58 | 93.62 | 87.18 | 99.58 | **99.60** |
| | C2 | 95.56 | 93.57 | 59.47 | 99.59 | 97.07 | 99.74 | 97.81 | 88.33 | 99.72 | **99.87** |
| | C3 | 71.87 | 52.08 | 47.39 | 94.50 | 92.13 | 95.77 | 98.76 | 45.92 | 98.92 | **99.51** |
| | C4 | 77.88 | 63.23 | 97.40 | 98.22 | 94.14 | 94.71 | 89.09 | 63.76 | 96.83 | **97.81** |
| ICA | C5 | 98.17 | **100.00** | 99.28 | 99.05 | 99.78 | 99.90 | 99.64 | 99.46 | 99.64 | 99.97 |
| | C6 | 70.47 | 56.20 | 79.35 | 93.05 | 99.35 | 98.93 | **99.90** | 46.75 | 99.48 | 99.68 |
| | C7 | 58.77 | 44.83 | 94.65 | 94.40 | 99.82 | 96.69 | 99.75 | 57.27 | 99.34 | **99.93** |
| | C8 | 85.38 | 71.20 | 79.43 | 92.24 | 93.55 | 94.00 | **99.74** | 51.03 | 96.88 | 98.13 |
| | C9 | 99.91 | 95.00 | **99.97** | 99.88 | 93.55 | 91.94 | 94.73 | 92.19 | 98.52 | 99.62 |
| OA | | 86.79 | 76.87 | 69.20 | 97.52 | 96.17 | 97.99 | 97.14 | 70.23 | 99.15 | **99.49** |
| AA | | 83.70 | 74.37 | 80.15 | 96.67 | 95.90 | 96.58 | 97.00 | 70.21 | 98.77 | **99.35** |
| Kappa | | 82.62 | 70.33 | 61.72 | 96.67 | 94.88 | 97.31 | 95.95 | 61.22 | 98.86 | **99.32** |
| *t* | | 4.22 | 92.92 | 473.09 | 97.94 | 4.49 | 39.17 | 189.01 | **2.19** | 3.78 | 3.97 |
| *tt* | | 3.01 | 1.84 | 50.63 | 17.23 | 2.48 | 3.67 | 7.21 | **1.05** | 1.08 | 1.31 |

(1) Compared with the conventional classification method SVM—the effects of BLS approximate to those of SVM methods on the Indian Pines and Salinas datasets. However, when BLS and SVM make use of the HSI data filtered by the Gaussian filter, the performance of GBLS was obviously better than that of GSVM. In the Pavia University dataset, the OA of BLS was 16.56% lower than that of SVM. After filtering the Pavia University data using the Gaussian filter, the OA of GBLS was about 3% higher than that of GSVM. SSBLS had the best performance. From Tables 3–5, the experimental results illustrate that the combination of the Gaussian filter and BLS contributes to improving the classification accuracy.

(2) HiFi-We firstly extracts different spatial context information of the samples by HGF, which can generate diverse sample sets. As the hierarchy levels increased, the pixel spectra features tended to be smooth, and the pixel spatial features were enhanced. Based on the output of HGF, a series of classifiers could be obtained. Secondly, the matrix of spectral angle distance was defined to measure the diversity among training samples in each hierarchy. At last, the ensemble strategy was proposed to combine the obtained individual classifiers and mSAD. This method achieved a good performance. But its performance in terms of OA, AA, and Kappa were not as good as these of SSBLS. The main reasons are that SSBLS adopts the advantages of spectral-spatial joint features sufficiently in the three operations of the Gaussian filter, BLS, and guided filter; these are useful to improve the accuracy of SSBLS.

(3) SSG assigns a label to the unlabeled sample based on the graph method, integrates the spatial information, spectral information, and cross-information between spatial and spectral through a complete composite kernel, forms a huge kernel matrix of labeled and unlabeled samples, and finally applies the Nyström method for classification. The computational complexity of the huge kernel matrix is large, resulting in increasing the consumed time of the classification. On the contrary, SSBLS not only has higher OA than SSG, but also takes lesser time than SSG.

(4) The EPF method adopts SVM for classification, constructs the initial probability map, and then utilizes the bilateral filter or the guided filter to collect the initial probability map for improving the final classification accuracy. The results of it were very good in the real three hyperspectral datasets. However, SSBLS had better performance compared with EPF. This is mainly because SSBLS firstly utilizes the Gaussian filter to extract the inherent spectral features based on spatial information, moreover, applies the guided filter to rectify the misclassification pixels of BLS based on the spatial context information.

(5) IFRF divides the HSI samples into multiple subsets according to the neighboring hyperspectral band, then applies the mean method to fuse each subset, finally makes use of the transform domain recursive filtering to extract features from each fused subset for classification using SVM. This method works very well. But the performance of SSBLS was better than that of IFRF. Specifically, the mean OA of SSBLS was 1.03% higher than that of IRRF in the Indian dataset, 0.24% higher in the Salinas dataset, and 1.5% higher in the Pavia University dataset. There were three reasons for the analysis results. Firstly, when SSBLS used the Gaussian filter to smooth the HSI spectral features based on the spatial information, the weight of each neighboring pixel decreased with the increase of the distance between it and the center pixel in the Gaussian filter window. The Gaussian filter operation could remove the noise. Secondly, in the SSBLS method, the integration of the Gaussian filter and BLS contributed to extracting the sparse and compact spectral features fusing the spatial features and achieved outstanding classification accuracy. Thirdly, SSBLS applied the guided filter based on the spatial context information to rectify the misclassified hyperspectral pixels for improving the final classification accuracy.

(6) The LPP_LBP_BLS method uses LPP to reduce the dimensionality of HSI in the spectral domain, then utilized LBP to extract spatial features in the spatial domain, and finally makes use of BLS to classify. The performance of LPP_LBP_BLS was very nice. But it has two disadvantages. First, the LBP operation led to an increase in the number of processed spectral-spatial features greatly. For example, the number of spectral bands after dimensionality reduction of each pixel was 50, and the number of each pixel spectral-spatial features after the LBP operation was 2950. Second, LPP_LBP_BLS worked very well on the Indian Pines and Salinas datasets, but the mean OA only reached 97.14% in the Pavia University dataset. It indicates that this method has a certain data selectivity and is not robust enough. The average OAs of SSBLS in the three datasets are all above 99.49%. In the Indian dataset, the mean OA is 99.83%, and the highest OA we obtained during the experiments is 99.97%. In the Salinas dataset, the average OA is 99.96%, and the highest OA can reach 100% sometimes. It shows that the robustness of SSBLS is better, especially on the Pavia University dataset. As the parameters change, the OAs change regularly, as shown in Figures 5c and 6c.

(7) Compared with BLS and GBLS. It can be seen in Tables 3–5 that BLS had an unsatisfactory classification effect only using the original HSI data; however, when the GBLS adopted the spectral features smoothed by the Gaussian filter, its OA was greatly improved. It indicates that the combination of the Gaussian filtering and BLS contributed to the improvement of classification accuracy. The classification accuracy of SSBLS was higher than those of BLS and GBLS. This was because SSBLS applied the guided filter based on the spatial contextual information to rectify the misclassified pixels, further improving the classification accuracy.

In summary, using the three datasets, the OA, AA, and Kappa of SSBLS were better than those of nine other comparison methods, as can be clearly seen from Figures 10–12. From Tables 3–5, it can be seen that the execution time of SSBLS was lesser than these methods (SVM, HiFi-We, SSG, EPF, GSVM, IFPF, and LPP_LBP_BLS), and the pretreatment time and the training time of SSBLS was lesser than HiFi-We, SSG, EPF, IFPF, and LPP_LBP_BLS.

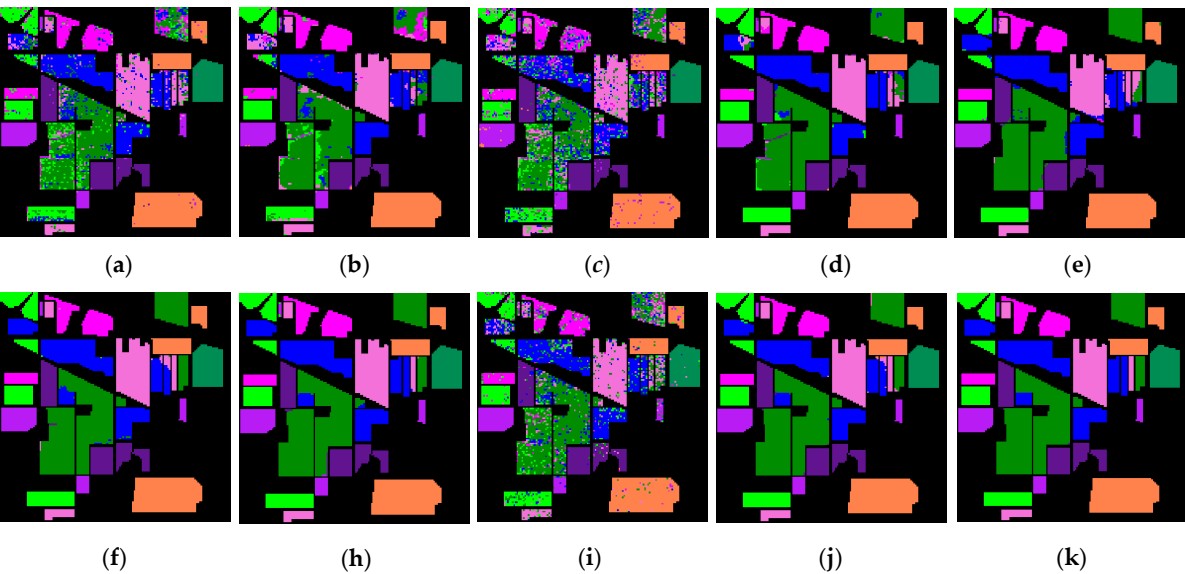

**Figure 10.** Classification maps of the Indian Pines dataset. (**a**) SVM 80.68%; (**b**) HiFi-We 86.88%; (**c**) SSG 69.14%; (**d**) EPF 95.83%; (**e**) GSVM 95.95%; (**f**) IFRF 98.86%; (**h**) LPP_LBP_BLS 99.74%; (**i**) BLS 78.58%; (**j**) GBLS 99.39%; (**k**) SSBLS 99.97%.

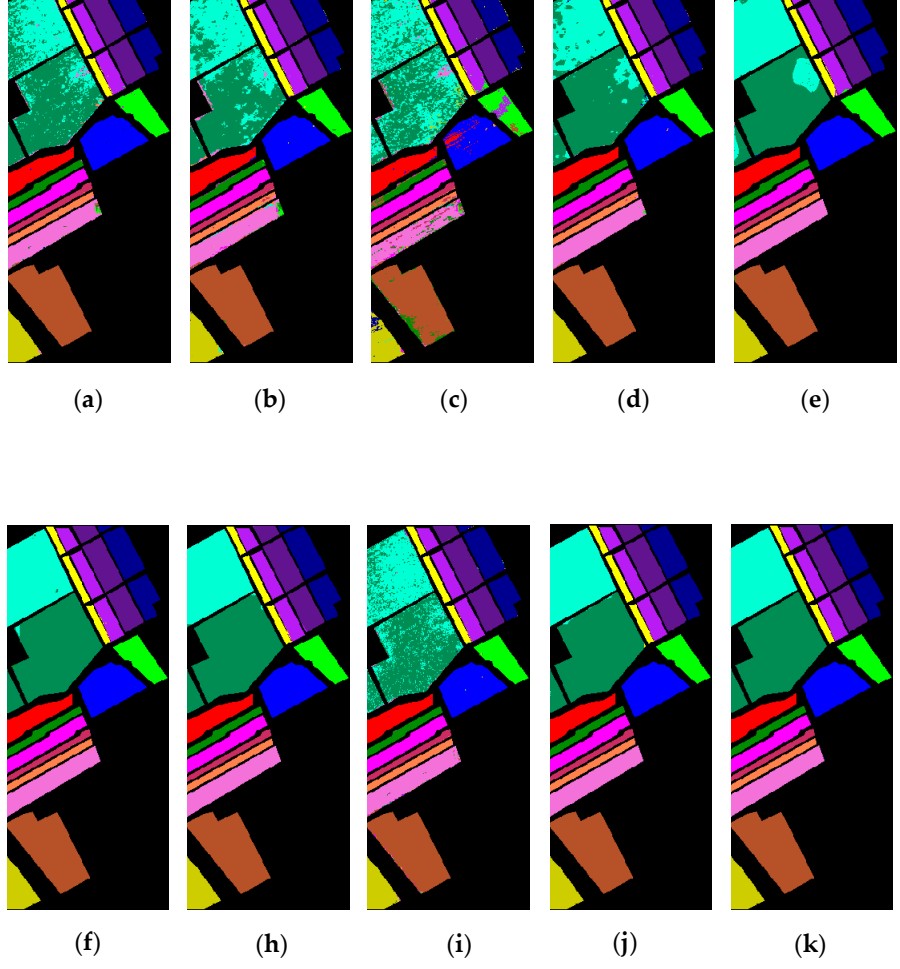

**Figure 11.** Classification maps of the Salinas dataset. (**a**) SVM 91.89%; (**b**) HiFi-We 91.10%; (**c**) SSG 81.46%; (**d**) EPF 95.92%; (**e**) GSVM 96.93%; (**f**) IFRF 98.86%; (**h**) LPP_LBP_BLS 99.83%; (**i**) BLS 92.00%; (**j**) GBLS 99.84%; (**k**) SSBLS 99.99%.

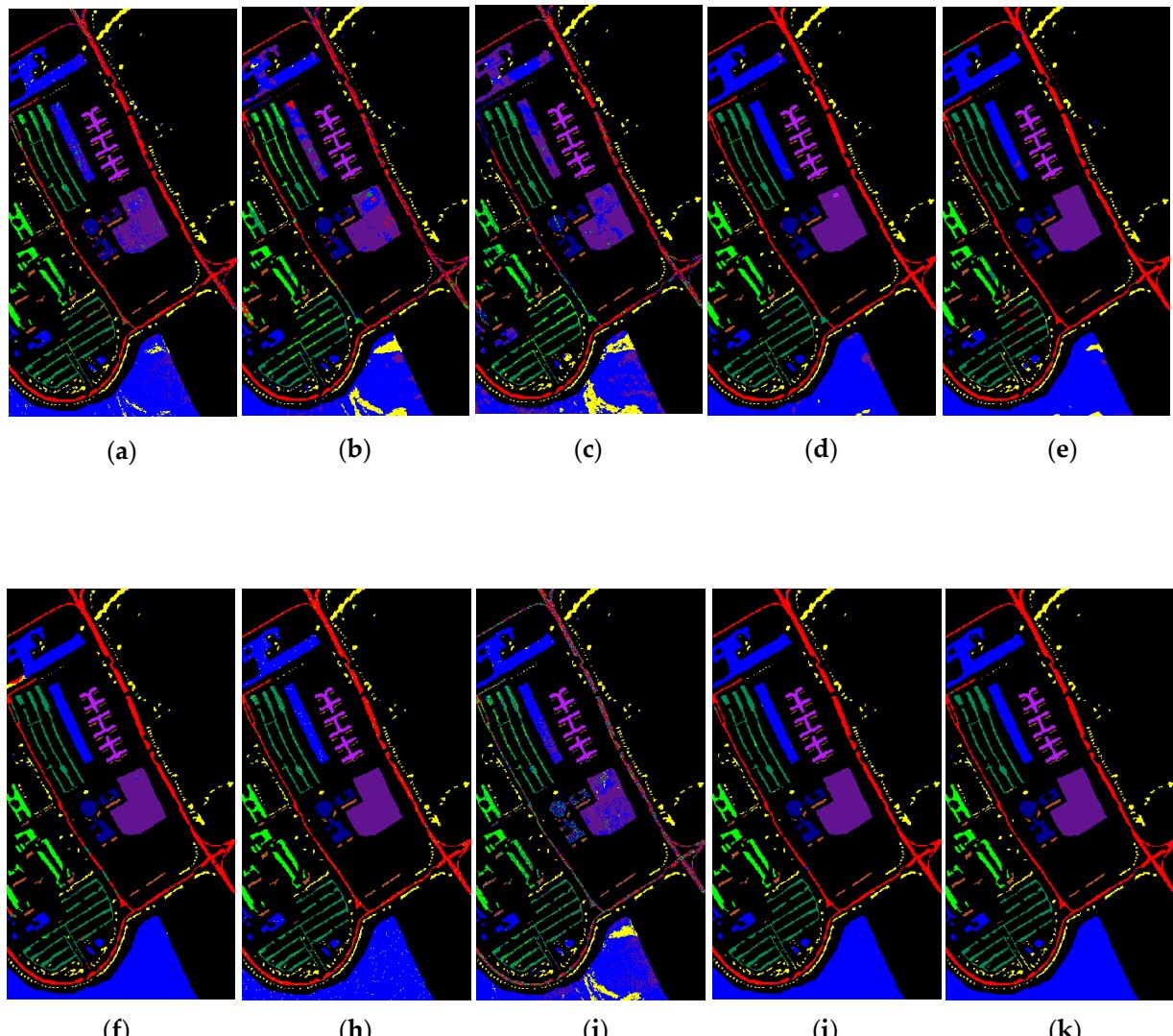

**Figure 12.** Classification maps of the Salinas dataset. (**a**) SVM 91.89%; (**b**) HiFi-We 91.10%; (**c**) SSG 81.46%; (**d**) EPF 95.92%; (**e**) GSVM 96.93%; (**f**) IFRF 98.86%; (**h**) LPP_LBP_BLS 99.83%; (**i**) BLS 92.00%; (**j**) GBLS 99.84%; (**k**) SSBLS 99.59%.

## 4. Discussion

The experimental results of the three public datasets indicate that SSBLS had the best performance in terms of three measurements (OA, AA, and Kappa) in all the compared methods. There were three main reasons for this, as follows. Firstly, the combination of the Gaussian filter and BLS contributed to the improvement of SSBLS classification accuracy. The Gaussian filter could fuse spectral features and spatial features of HSI effectively to extract the inherent spectral characteristics of each pixel. BLS expressed the smoothed spectral information into the sparse and compact features in the process of mapping feature using random weight matrixes fine-turned by the sparse auto encoder. It also improved the classification accuracy. It can be clearly seen from Tables 3–5 that the performances of GBLS and SSBLS using the HSI data smoothed by the Gaussian filter were greatly improved. Secondly, SSBLS takes full advantage of spectral-spatial joint features to improve its performance. The Gaussian filter firstly smooths each band in the spectral domain based on the spatial information to achieved the first fusion of spectral and spatial information. The guided filter corrects the results of BLS classification under the guidance of the grey-scale guidance image, which is obtained by the first PCA based on the spectral information from the original HSI. These operations join spectral features and

spatial information together sufficiently. At last, SSBLS applies the guided filter to rectify the misclassification HSI pixels to further enhance its classification accuracy.

## 5. Conclusions

To take full advantage of the spectral-spatial joint features for the improvement of HSI classification accuracy, we proposed the method of SSBLS in this paper. The method is divided into three parts. Firstly, the Gaussian filter smooths each spectral band to remove the noise in spectral domain based on the spatial information of HSI and fuse the spectral information and spatial information. Secondly, the optimal BLS models were obtained by training the BLS using the spectral features smoothed by the Gaussian filter. The test sample labels were computed for constructing the initial probability maps. Finally, the guided filter is applied to rectify the misclassification pixels of BLS based on the HSI spatial context information to improve the classification accuracy. The results of experiments of the three public datasets show that the proposed method outperforms other methods (SVM, HiFi-We, SSG, EPF, GSVM, IFRF, LPP_LBP_BLS, BLS, and GBLS) in terms of OA, AA, and Kappa.

This proposed method is a supervised learning classification that requires more labeled samples. However, the number of HSI labeled samples were very limited, and a high cost is required to label the unlabeled samples. Therefore, the next step is to study a semi-supervised learning classification method to improve the semi-supervised learning classification accuracy of HSI.

**Author Contributions:** All of the authors made significant contributions to the work. G.Z. and Y.C. conceived and designed the experiments; G.Z., X.W., Y.K., and Y.C. performed the experiments; G.Z., X.W., Y.K., and Y.C. analyzed the data; G.Z. wrote the original paper, X.W., Y.K., and Y.C. reviewed and edited the paper. All authors have read and agreed to the published version of the manuscript.

**Funding:** This research was funded by the National Natural Science Foundation of China under Grant 61772532, Grant 61976215, and Grant 61703219.

**Data Availability Statement:** Publicly available datasets were analyzed in this study. This data can be found here: http://www.ehu.eus/ccwintco/index.php?title=Hyperspectral_Remote_Sensing_Scenes.

**Conflicts of Interest:** The authors declare no conflict of interest.

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
