# Peer review of "Spectral-Spatial Joint Classification of Hyperspectral Image Based on Broad Learning System"

_remotesensing, doi:10.3390/rs13040583_

Round 1

Reviewer 1 Report

  1. For spectral dimension reduction of HSI, other than feature extraction, feature selection and band selection methods can be adopted in supervised, unsupervised, semi-supervised and weakly supervised ways.  Hence, those works should be literatured.
  2. References are required for your statement of "LPP is widely used in the spectral feature extraction of HSIs". 
  3. Why use ICA instead of LPP in your BLS?
  4. Second paragraph of part one, I don't see the logical connection between the last two sentences with the former sentences about feature extraction.
  5. As one of the most popular spatial feature extraction methods, literature review for mathematical morphology based methods and solutions for potential shortages are too weak, e.g., MP with partial reconstruction, object based MP, maximal statable extreme region guided MPs etc.
  6. Literature review for applications of BLS is weak.
  7. Literature review enhancement is recommended for the applications of Gaussian filter.
  8. Check your variables used in text lines, some of them were not in the line center.
  9. Check your equation and keep them in the same size
  10. Figure 2, Check your legends, font type seems not the right form, and "bitumen" should be "Bitumen".
  11. Original color code usage for experimental datasets and classification maps is recommended.
  12. Since the authors announced that their goal is to address the issues of "many methods of processing spectral characteristics and spatial features often take up lots of storage space or require numerous calculation", experimental results should be provided to strongly support their announcement.
  13. Figure charts are large indeed, but the words and numbers in it are too small, try to solve this issue.
  14. Check your abbreviations usage, some were defined repeatedly, some were not.

Reviewer 2 Report

  1. The motivation of this paper is to address the tremendous storage space and numerous calculations of current HSI classification methods. However, the main efforts of this paper aim to enhance the accuracy in the abstract. It is confusing that how the proposed method tackles the storage and efficiency issues.
  2. In the experiments, it is nice to report the execution time. On all three datasets, every method runs quite fast. The most time-consuming on the largest dataset is around 1300s, which is acceptable. Therefore, the efficiency is not a big issue.
  3. There is no report on the storage space. 
  4. The motivation and the proposed methods are not matchable. Therefore, it is prohibited to evaluate the technical contributions. I would like to suggest the authors to clearly illustrate the motivation/challenges and how the proposed method deal with these challenges.

Reviewer 3 Report

This paper proposed a spectral-spatial joint model for HSI classification. In their model Gaussian filter and broad learning system are integrated together to generate the initial probability map on the spectral domain. Then the guided filter is applied to correct the probability map on the spatial domain. Finally, the classification results can be obtained according to the maximum probability principle.  In general, the paper is well written and easy to follow. The experimental results are also solid.

However, I still have the following concerns:

The contribution of this paper hasn’t been clearly demonstrated.

To verify the advantage in computation cost, the authors should show a comparison of the test time between different algorithms.

In page 2, authors claim that the integration of Gaussian filter and BLS has strong feature expression ability. It is worth showing feature maps and making detailed illustration.

Round 2

Reviewer 1 Report

No more comments.

Reviewer 2 Report

Thanks for the revision. Here are my new comments.

  1. The authors directly provide their solution; however, why their solution can tackle the challenges is missing. More explanation on the philosophy in Line 110-119 is needed.
  2. Eq. (5) looks weird to me. What does the brackets mean?
  3. A notation table is suggested.
  4. The visualization of Figure 7-9 can be improved. There could be several lines in one figure.
  5. An ablation study on different components is suggested, which helps the analysis on which part brings in the improvements.
